



**Seasonal Sinking rates of Transparent Exopolymer Particles (TEP) concentrations with**
**associated Carbon flux in adjacent Bohai Sea and Yellow Sea**
M Shahanul Islam[1,2,3], Sun Jun[2,3*], Li Xiaoqian[2,3], Leng Xiaoyun[2,3]
1. Collegeof Food Engineering and Biotechnology, Tianjin University of Science and
Technology University, Tianjin 300457, China
2. Tianjin Key Laboratory of Marine Resources and Chemistry, Tianjin University of
Science and Technology, Tianjin 300457, PR China
3. Research Centre for Indian Ocean Ecosystem, Tianjin University of Science and
Technology, Tianjin 300457, China
*Corresponding Author: phytoplankton@163.com
**Abstract**
To study the seasonal transparent exopolymer particles (TEP) distributions,
sedimentation and its impacts on carbon cycle in north Chinese seas, a total of total 56
stations TEP samples and its sinking rate measurements by SETCOL method via water
sampling cruise during autumn (2014), summer (2015) and winter (2015) in the Bohai Sea
(BS), North Yellow Sea (NYS) and South Yellow Sea (SYS) at three different depths were
carried out. Temperature, phytoplankton, chlorophyll-a (Chl-a) and salinity with five
nutrients, phosphate (DIP), silicate (DSi), dissolved inorganic nitrate (DIN) (including nitrite,
nitrate and ammonium) were also collected and measured for correlation analysis to visualize
the seasonal effects on TEP concentrations (CTEP) and its sinking. Average of total CTEP
(2.13 μg Xeq L$^{-1}$) was higher in NYS (3.32 μg Xeq L$^{-1}$) costal currents with highest average
CTEP during winter (6.17 μg Xeq L$^{-1}$) specially in NYS (7.00 μg Xeq L$^{-1}$) through coastal
current mixing zone. Average of total sinking rates (1.03 mD$^{-1}$) was higher in SYS (1.09
mD$^{-1}$) through mid-water layer than other seas, especially in autumn (1.13 mD$^{-1}$) with higher
seasonal average sinking rates at summer (1.04 mD$^{-1}$). Carbon associated with TEP (TEP-C)
was averagely distributed (1.47 μg C L$^{-1}$) at subsurface layer of study areas. Seasonal highest
distribution of TEP-C was 4.44 μg C L$^{-1}$ during winter, mostly in NYS. Dominant
phytoplankton species *Paralia sulcata, Thalassisira excentrica* and *Rhizosolenia styliformis*
maintained average correspondences with CTEP which may indicate the influences of them
on TEP concentration. Congregating oceanic stations in other groups, coastal stations were
averagely clustered together in multivariate analysis. Average canonical correspondence



analysis showed close relation of CTEP with Chl-a during autumn and with nutrient during
winter.
**Keywords**: Transparent Exopolymer Particles, sinking rate, carbon sink, seasonal variation,
Bohai Sea, Yellow Sea.
**1 Introduction**
Transparent exopolymer particles (TEP) are macro-gels like substances that play an
active role in the marine carbon cycle between particulate and dissolved organic carbon (POC
and DOC, accordingly), by extending the size continuum, in addition to assisting particle
formation (Alldredge et al., 1993; Passow, 2002b; Verdugo et al., 2004). TEP are generally
consider as transparent particles which can be stainable by Alcian Blue, a dye that favorably
binds to acidic polysaccharides after complexing with carboxyl groups and sulfate (Alldredge
et al., 1993; Passow and Alldredge, 1995). Abiotically TEP sourced from dissolved
polysaccharides secreted by phytoplankton (Logan et al., 1995; Passow, 2002b; Thuy et al.,
2015). As TEP possess surface-active characteristics with neutral buoyancy, they are
scavenged easily with gas bubbles and aggregated at the sea surfacelayer which can make sea
surface microlayer (SML) organically (Azetsu-Scott and Passow, 2004; Cunliffe et al., 2013;
Mopper et al., 1995; Wurl et al., 2009).
A combined effect of regional biological and physical factors, including salinity, air-
driven turbulence and production of dissolved polysaccharide primer by phytoplankton and
bacteria can controlled the formation and distribution of TEP. Nutrient ranges control
phytoplankton community, which is one of the relative factors that thrust the partitioning of
organic matter between particulate and dissolved phases (Carlson et al., 1998; Conan et al.,
2007; Lomas and Bates, 2004; Thornton, 2014), and organic matter and production of TEP
(Claquin et al., 2008; Corzo et al., 2000; Mari et al., 2005; Passow, 2002a). Relationships
between TEP and chlorophyll *a* (Chl-a) develop during a bloom, resembles the production of
TEP by phytoplankton (Passow, 2002b). This relationship during bloom phase is species-
specific, with the cellular aggregation rate of TEP by phytoplankton caused by their growth
period. So, if the area is composed with other phytoplankton taxa with different life stages,
the relation will not match with special gradient between TEP–Chl-a. Low nutrient increases
TEP abundances through preventing TEP consumption by bacteria (Bar-Zeev and Rahav,
2015), so that aggregation and accumulation of TEP at sea surface as N-poor C-rich organic
material (Passow, 2002b).





In the sea surface microlayer (SML), gel particles formations and abundance have been observed (Orellana et al., 2011; Wurl et al., 2011b) which resembles the relation of gels with carbon cycling at the surface layer. Surface active TEP is transported to the SML after accumulation with rising bubbles derived from wave's action (Wurl et al., 2011a; Wurl and Holmes, 2008). Although these actions distribute its substances into subsurface layer temporarily though SML has a rapid reformation intensity (Cunliffe et al., 2013). Biogenic material concentrated in the SML may also be mixed with atmosphere through bubble rupturing which may capable of ice nucleation and cloud condensation (Bigg and Leck, 2008; DeMott et al., 2015; Orellana et al., 2011; Quinn et al., 2014; Wang et al., 2015; Wilson et al., 2015). Since TEP sticks to elevated bubbles (Mopper et al., 1995) and are accumulated in the SML, TEP may contribute to the organic increment of sea spray aerosols produced from film droplets (Aller et al., 2005).

TEP are more adherer than non-TEP substances which may help in particles and subsequently increase sedimentation (Logan et al., 1995; Passow et al., 1994). As microbial hotspots, TEP and POC serve a source of carbon to the deep ocean by sinking in water column. Although sinking process of marine aggregates have been observed (Burd and Jackson, 2009; Iversen and Robert, 2015; Jokulsdottir and Archer, 2016; Prairie et al., 2015), there remains the necessity of understanding about the effect of TEP in the biological carbon pump (Burd et al., 2016; Zetsche and Ploug, 2015). TEP sinking rate measurement experiments in Changjinag (Yangtze River) estuary near EastChina Sea during summer showed higher sedimentation of TEP at upper water layer than deep seas (Guo & Sun 2018). It also showed higher sinking rates of TEP in summer rathe then spring which may have important role about carbon exports in that area.

Present study was conducted through the semi-closed Bohai Sea (BS) and Yellow Seawhich covered a big part of the north Chinese seas. Semi enclosed BS is located at the north-eastern continental region of China (Xu et al. 2010). With sensitive primary productivity and commercial fishery (Tang et al. 2003, Huang et al. 1999), BS got waste water-loads and inputs from the Tianjin City as well as Liaoning, Shandong and Hebei provinces (Xu et al. 2010). At eastern of BS, there is another semi-enclosed marginal sea of the western Pacific Ocean which is the Yellow Sea (Liu et al. 2015b). Yellow Sea possesses various oceanic processes through seasons (Hwang et al. 2014; Su 1998; Yuan et al. 2008; Isobe 2008). Visualizing better seasonal scenario, oceanic area of Yellow Sea was divided into two major water body (Li et al. 2017) i.e. North Yellow Sea (NYS) and South Yellow





Sea (SYS).In autumn, appurtenance of Yellow Sea cold water mass(YSCW) at SYS influenced the vertical mixing of NYS water (Zou et al. 1999). The Yellow Sea Warm Current (YSWC) acted at SYS during winter (Gao et al., 2004, Su 1998) and Changjiang Diluted water (CDW) during summer (Naimie et al. 2001). Fishery (Tang and Su, 2001), biological community (Hyun and Kim, 2003; Fu et al., 2009; Zhang et al., 2009) and ecological problems raises the importance of researches on Yellow Sea (Sun et al., 2011; Tang et al., 2007, 2010).

Studies on the TEP and its sinking rates showed correlation with environmental parameters i.e. temperature (Claquin et al. 2008, Fukao et al. 2012), salinity (Mari et al 2012) and phytoplankton species composition (Passow 2002b). On the other hand, it also showed correlation with nutrients in different studies (Corzo et al. 2000, Mari et al 2005). Seasonal variation of parameters can be a cause behind these phenomena. Poor correlations among TEP and these environmental parameters can be found due to limited data and insignificant variations in salinity and nutrients from sampling stations (Guo & Sun 2018). Seasonal distribution and sinking rate of TEP will describe more specific relation of TEP with other environmental parameters. On the basis of these objectives, present study was conducted on sinking flux of TEP and its concentration with related carbon exports through three separate seasons (autumn, summer, winter) in the Bohai and Yellow Sea of China from 2014-2015.

## 2 Materials and Methods

### 2.1 Study area

Water sample (1 liter) was taken from three different depths (0-100 meters) at various stations (Fig. 1) during 2014-2015 in Bohai Sea (BS), North Yellow Sea (NYS) and South Yellow Sea (SYS). Autumn sampling was conducted between 8-22 November (Fig. 1A), 2014, summer sampling from 18 August to 22 September, 2015 (Fig. 1B) and winter sampling in Bohai Sea and North Yellow Sea from 17 October to 22 November, 2015 (Fig. 1C). Boundary currents flow directions in Bohai Sea and Yellow Sea were changed with seasons (Fig. 2). Korean coastal currents (KCC) flows northward in summer and southward in winter. YSWC was found during winter and CDW flows eastward during summer. Remain costal currents and warm currents flows in their constant direction (Hwang et al. 2014; Su 1998; Yuan et al. 2008; Isobe 2008) which may have distributary effect on these particles concentration and its sinking rates besides all hydrological parameters.



Sampling stations (St.s) of autumn through the Bohai Sea (St.s B45, B49, B57, B62,
B68), North Yellow Sea (St.s B04, B11, B15, B19, B25, B34) and South Yellow Sea (St.s
H07, H09, H10, H18, H20, H26, H27, H33, H35, H40) were determined by geographical
positions. Similarly, the stations of Bohai Sea (St.s B45, B49, B57, B62, B67, B68), North
Yellow Sea (St.s B04, B10, B15, B19, B25, B34) and South Yellow Sea (St.s H07, H09,
H10, H18, H19, H26, H28, H33, H35, H38) were associated accordingly in summer. In
winter, sampling was done only in Bohai Sea (St.s B42, B45, B47, B62, B68, BS1, BS5) and
North Yellow Sea (St.s B04, B08, B15, B22, B25, B33), except SYS.

**2.2 Sample collection**

Sample collection was done by multiple rosette (with CTD sensors) for different
depths at each sampler stations based on bottom depth. Each station wat designed with three
distinguishing depths for better graphical analysis. Samples were collected to determine
phytoplankton composition, chl-a concentration, TEP abundances and its sinking rates
nutrients separately. Due to shallow water, depths were limited in each station between 0-100
meters. In 1L sampling bottle, phytoplankton samples were collected with 5% formaldehyde
concentration for further analysis (Guo and Sun 2018). For Chl-a analysis (Chl-a S), gathered
sea water of each sampling depths was filtered through 25 mm GF/F and stored in -20 ºC. Sea
water were collected in 100ml sample bottle from all sampling depths and stored at -25ºC for
nutrient analysis of each station. CTD sensors recorded temperature and salinity was
determined while sampling from different depths from study area.

**2.3 Biological parameters**

According to Welschmeyer (1994), chlorophyll – a (chl a) were measured after
filtering seawaters from all stations by 25 mm GF filters. Chl a concentration in samples
determined after were filtered onto 25 mm GF/F filters (Whatman[TM]) and then reserved at -
20°C in the dark until analysis. 90% acetone were used for extraction of chl-a for 24 h at -
20°C in the dark, and samples were then analyzed by a laboratory fluorometer called Turner-
Designs Trilogy[TM]. Phytoplankton sample (1 Liter, preserved with 30% formaldehyde) were
analyzed according to modified Utermöhl methods followed by Sun et al. (2002) after
arranging the samples (25 ml) in Utermöhl counting chamber (being settled for 24 hrs.) in
inverted microscope.

**2.4 Chemical analysis**





Nutrients i.e. dissolve inorganic phosphate (DIP), dissolve inorganic nitrogen (DIN),
dissolve silicates (DSi), ammonium, nitrate & nitrite analysis were measured by fully
automated (SANPLUS, Dutch SKALAR company) wet chemical analyzer (Liu et al., 2015a).
Measurement of TEP were sextuplicate for all samples from sampling depths by following
colorimetric method of Passow and Alldredge (1995) after confirming xanthan gum curve by
absorption measurement. At least 50ml ($V_f$) sample sea water were thoroughly filtered (4-6
replicates) at low and fixed vacuum (150 mm of Hg) through polycarbonate filters (0.4-pm
pore-size) and dye binding particles on the filter for approximate 2 seconds with 500 micro
liters of 0.02% alcian blue (8GX; aqueous solution) in 0.06% acetic acid (pH 2.5). After
staining, filters are rinsed carefully with distilled water to prevent excess dye once. Dye
bound to substrates will not wash off by this rinsing. Filters are then soaked into 25-ml
beakers with 6 ml of 80% $H_2SO_2$ and kept for 2 hours. The beakers should be gently shacked
for 3-5 times during soaking period. Maximum absorption of the solution ($E_{787}$) lies at 787
nm and it was measured (μg Xeq $L^{-1}$) in a l-cm cuvette ($B_{787}$) against distilled water as a
reference. The equation is:

174                  CTEP = ($E_{787}$-$B_{787}$) × $(V_f)^{-1}$×fx

Where, fx=Average calibration factor and it was 9.83 μg from the graph of xanthan gum
curve.
**2.5 Measurement of TEP sinking flux**
TEP sinking rates were determined at each station, and the SETCOL method was used
to measure the sinking rates according to Bienfang (1981). For measurements, a Plexiglass
column (height = 0.45 m and volume = 750 ml) was filled completely with a homogeneous
water sample within 10 min after sampling, and a cover was then placed on the set-up. In the
vessel, the Plexiglass column was kept to settle undisturbed for 2-3 hours, and a
thermostatically controlled water bath with water jackets controlled the temperature was
maintained by pumping its water. The settled sample of experiment was collected in sample
bottles by successively draining the upper, middle, and bottom compartments of the
Plexiglass column via piped outlet in the wall of column. The TEP biomass was measured
after the settlement in the samples from all three compartments. These measurements were
combined to calculate the sinking rate of TEPs according to the formula:

$$V = \frac{B_s}{B_t} \times \frac{L}{t}$$






where V = sinking rate; $B_s$ = the biomass of TEP settled into the bottom compartment; $B_t$ =
the total biomass of TEPs in the column; L = length of the column; and t = settling interval.
Samples from all depts were triplicated during measurement for better data analysis and
marked according to stations about sinking rate as well as TEP concentrations.
**2.6 Data analysis**
Study stations in the map during autumn 2014 (Fig. 1A) and summer 2015 (Fig. 1b)
were sectioned in three vertical view for better understanding of the sample concentrations.
Stations in the map of winter 2015 (Fig. 1C) was divided according to seas (Fig. 1D).
Seasonal currents flow maps (Fig. 2) were built on the basis of secondary data and previous
literatures (Hwang et al. 2014; Su 1998; Yuan et al. 2008; Isobe 2008) from that area.
Analysis and discussions were forwarded according to both seasonal (autumn, summer &
winter) and oceanic (Bohai Sea, North Yellow Sea & South Yellow Sea) categories (Table 1).
Calculation of TEP-carbon (TEP-C, $\mu g\ C\ L^{-1}$) was determined with the slope (0.75) from the
equation as follows (Engel &Passow 2001):

TEP-C = 0.75 × TEP$_{color}$ (Guo & Sun 2018)

where TEP$_{color}$ is the TEP concentration (CTEP) with the unit of $\mu g\ Xeq\ L^{-1}$.
Various multivariate analyseswere performed by using Multi Biplots software (Vicente
Villardón, 2015) on recorded data. Single cluster analysis was performed by Multivariate
Statistical Package Software with Baroni-UrbaniBuser Coefficient. Linear regression,
Pearson correlation and covariance were performed by Microsoft Excel 2016 software.
Canonical correspondence analysis (CCA) were done by Canoco software, version 4.14
(CANOCO for Windows; Ter Braak&Šmilauer, 2002).Dominance index was used to
describe phytoplankton dominant species under this equation:

$Y = \dfrac{n_i}{N} \times f_i$

Where, N is the total cell abundance of all species, $n_i$ is total cell of species $i$ and $f_i$ is the
count of occurrence of species $i$ in all sample (Guo et al. 2014). For integrated surface view
of recorded and examined parameters during winter 2015, Surfer 12 was used. Box-whisker
plots by Microsoft Excel 2016 showed the range of all recorded parameters after integration.
Concentrations of different parameters were graphically presented by Ocean Data View
(ODV 2016) software.






## 3 Results

### 3.1 Environmental Hydrology

The Bohai Sea and Yellow Sea had a complex dynamic environment with various
seasonal and local geophysical currents (**Fig**. 2). Average concentration of all parameters
through every season showed high chl-*a* concentration along coastal zones of BS and NYS.
Average annual CTEP was higher at BS along BSCC, at NYS along YSCC and at SYS along
CDW. Average annual nutrients were highly concentrated at BS than other seas, except
nitrite at YSCC of SYS (Table 1).

During autumn 2014 (Fig. 2C), CTEP was higher at the north of NYS along LCC and
CWC of southern SYS. Temperature were higher at YSCC of SYS and salinity were dense at
KCC of NYS and SYS with high Chl-*a* was at LCC of NYS. Concentrations of nutrients
were higher at BS except nitrite. Nitrite was higher at LCC of NYS and at YSCC of SYS.
DIP. DIN, DSi and nitrate were higher at the southern part of SYS through whole autumn
(Table 1).

During summer 2015 (Fig. 2A), CTEP was higher at southern SYS (Fig 3) with high
temperature and nutrients (DIP, DIN, DSi and nitrates). Notably, DIP found high at CWC of
SYS. In BS, nitrate and nitrite were higher at south coast and ammonium was higher at north-
west coast with high temperature. Salinity was aggregated at KCC of NYS and mid SYS.
Chl-*a* was dense at YSCC of NYS and CDW of SYS (Table 1).

During winter (Fig. 2B), CTEP was dense at KCC of NYS with high temperature and
salinity. However, chl-*a* was higher at the north coast and BSCC of BS with YSCC of NYS
too. Most of the nutrients were higher at BSCC of BS except DIP. DIP found higher at the
transitional zone of BS and NYS.DSi concentration was also higher at KCC of NYS.

### 3.1.1 Vertical concentrations in autumn 2014

Vertical profile showed higher concentration of CTEP at SYS with high temperature
too, especially at St. H33, H35 and H40. Chl-a was dense in NYS (St. B15, B19 and B25)
where DSi was dramatically low. Nutrients i.e. DIN, nitrate and ammonium were found
higher at the bottom of BS, especially at St. B45, B49 and B57. Salinity was lower at surface
of whole study area. DIP found lowest at surface stations i.e. H07 and H09 of SYS but



highest at bottom of B34 and H07 stations. Nitrite was found higher at the SCM of NYS and
SYS (Table 1). During autumn, dominant phytoplankton were *Paraliasulcate, Coscinodiscus*
sp., *Ceratiumfusus, Thalassiosira sp., Probosica alata f. indica, Ceratiumtripos, Nitzschia*
sp., *Thalassiosira pacifica, Guinardia delicatula* and *Thalassiosira excentrica* sequentially.

### 3.1.2 Vertical concentrations in summer 2015

With obvious high temperature, summer possessed low CTEP at surface of BS. CTEP
was higher with low nutrients and chl-a at the SCM of Station B45. DIN, nitrate and nitrite
were high near station B68. DIP and DSi were higher at the bottom of Station B57 in BS.
CTEP was comparatively low at NYS with high chl-a at SCM and bottom of Station B25.
DIP, DSi and nitrate were higher at the bottom of Stations B04 and B11. *Alexandrium*
*tamarense, Rhizosolenia styliformis, Paralia sulcate, Guinardia flaccida, Dinophysis* sp.,
*Ceratium fusus, Thalassiosira excentrica, Ceratium furca, Dictyocha fibula* and *Diploneis*
*bombus* were dominant phytoplankton accordingly through study area in summer. In SYS,
CTEP was higher at the SCM with low nutrients and chl-a as BS. Nutrients along with
salinity were higher at the bottom of Station H-7 and H09.

### 3.1.3 Vertical concentrations in winter 2015

In winter, NYS showed higher abundances of Chl-a and salinity at the stations (B04,
B25 & BS5) near YSCC. Higher CTEP was found at SCM of NYS than BS. Temperature
was recorded lowest in BS than NYS (**Fig**. 1), especially near shore area. Phytoplankton i.e.
*Paralia sulcate,Thalassiosira excentrica, Actinoptychus* sp*., Donkinia recta, Thalassiosira*
sp., *Coscinodiscus* sp., *Coscinodiscus subtilis, Navicula* sp*., Pleurosigma pelagicum* and
*Coscinodiscus radiatus* were dominant during winter.  Nutrients i.e., DIN, DSi, ammonium,
nitrate and nitrite found higher at the SCM of Bohai Sea. DIP was found abundance in
surface area at the transitional area (near Station B33) of BS and NYS (Table 1). Average
nutrients were higher in BS than NYS during winter 2015. Stations near coastal area of BS
(B45 & B68) and NYS (B04, B25 & BS5) have higher nutrients comparatively.

### 3.2 Seasonal and regional TEP concentration

Present study measured 2.13 μg Xeq. L$^{-1}$ as average TEP concentration (CTEP) which
is ranged between 0.2-23.20 μg Xeq. L$^{-1}$. NYS (2.32 μg Xeq. L$^{-1}$) has more TEP
concentration in average than SYS (1.18 μg Xeq. L$^{-1}$) and BS (2.08 μg Xeq. L$^{-1}$). Apparently,
winter season (6.17 μg Xeq. L$^{-1}$) shows higher average concentration of TEP in each sea





(Table 2) than in summer (1.10 µg Xeq. L$^{-1}$) and in autumn (0.67 µg Xeq. L$^{-1}$). SYS showed
higher CTEP in autumn (0.93 µg Xeq. L$^{-1}$) and in summer (1.42 µg Xeq. L$^{-1}$) than NYS and
BS.

**3.3 TEP associated carbon abundances**

The carbon (TEP-C) associated with TEP picked at winter 2015 and became lower

during autumn 2014. BS showed low carbon abundance at surface during summer (Fig. 4E).
NYS possessed high TEP-C in SCM during winter (Fig. 4J) and at bottom during autumn and
summer (Fig. 4B & 4F). Surface of SYS recorded high TEP-C than BS and NYS during
summer (Fig. 4G). However, SYS also possessed comparatively high verity of TEP-C than
BS and NYS at its SCM during autumn and summer (Fig. 4D & 4H). Average TEP-C
showed high variation at SCM through study areas (Fig. 4L).

Average TEP-C was 1.47 µg C L$^{-1}$ with seasonal highest 4.44 µg C L$^{-1}$ during winter

specially in NYS (5.25 µg C L$^{-1}$). SYS has higher TEP-C during autumn (0.58 µg C L$^{-1}$) and
summer (1.07 µg C L$^{-1}$) than other seas. Highest TEP-C was found at SML of stations B15
(15.78 µg C L$^{-1}$), B22 (17.40 µg C L$^{-1}$) and B33 (10.91 µg C L$^{-1}$) of NYS during winter which
showed close cluster during analysis.

**3.4 Seasonal and regional TEP sedimentation**

Sedimentation or sinking rates of TEP was recorded 1.03 mD$^{-1}$ in average of all

seasons (Table 3) at study area. TEP sinking rate was higher in summer (1.04 mD$^{-1}$) than
autumn (1.02 mD$^{-1}$) and winter (1.03 mD$^{-1}$). However, winter showed high sinking rates
(1.02 mD$^{-1}$) in BS than its summer (1.01 mD$^{-1}$) and autumn (0.9 mD$^{-1}$) data. On the other
hand, comparatively high sinking rates were measured in SYS during autumn (1.13 mD$^{-1}$)
than its summer (1.05 mD$^{-1}$). SYS also possessed average high sinking rates of TEP (1.09
mD$^{-1}$) than BS (0.9 mD$^{-1}$) and NYS (1.03 mD$^{-1}$).

**3.5 TEP sinking in segmented depths**

Average TEP sinking dynamics were similarly close in average at each depth (Fig.

5K). In BS, high sedimentation variation was observed at mid layer (Fig. 5A) during autumn
(0.72-1.16 mD$^{-1}$) and at surface (Fig. 5E,5I) during summer (0.86-1.57 mD$^{-1}$) and winter
(0.64-2.06 mD$^{-1}$). Surface of NYS (Fig. 5B) has diverse sinking rates during summer (0.77-
2.47 mD$^{-1}$), mid layer (Fig. 5F) in winter (0.72-1.53 mD$^{-1}$) and bottom (Fig. 5J) in autumn
(0.76-1.19 mD$^{-1}$). Surface of SYS (Fig. 5C) during summer (0.62-2.40 mD$^{-1}$) and bottom



layers (Fig. 5G) during autumn (0.78-1.55 mD$^{-1}$) showed sedimentation variation
accordingly. In average, bottom layer (Fig. 5D) during autumn (0.72-1.55 mD$^{-1}$) and surface
of study area during summer (0.62-2.47 mD$^{-1}$) and winter (0.64-2.06 mD$^{-1}$) possessed high
sinking dynamicity (Fig. 5D, 5H & 5L).

**3.6 Correspondence relationships of TEP**

TEP showed close correspondent relationship with DIP and chl-*a* in average (**Fig. 6**I)
and in winter at BS (**Fig. 6**F) after applying CCA. However, winter at NYS showed TEP was
corelated with nitrate through each station. In autumn, TEP showed average close
correspondence with nitrite in CCA across study areas (**Fig. 6**A, 16D, 16G& 16J). During
summer, TEP was averagely compliance with nitrite (**Fig. 6**D) through all seas (**Fig. 6**H and
12I) except at BS (nitrate; **Fig. 6**E). In average, TEP showed close correspondences with *T.*
*excentrica* and *P. sulcata* during autumn*, R. styliformis* in summer and *P. sulcate* and
*Actinoptychus* sp. were dominant across study area. Both in BS and NYS, *P. sulcata* was
highly dominant and mostly correlates with concentration of TEP through all seasons. In
SYS, correspondences of dominant phytoplankton with TEP were observed rather than
nutrients (Fig. 6J& 6K).
Considering all parameters, most of the SYS stations clustered closely in dendrogram
(Group 1 and 4) after applying Baroni-Urbani Buser coefficient during autumn (**Fig**. 13A)
and summer (**Fig**. 13B). In BS, St. B45 clustered in same group with B68 through all season
(autumn; group 2, summer; Group 5 and winter; Group 7). St. B15 of NYS showed group
correspondence with St. B25 during autumn (**Fig**. 13A; Group 2) and winter (**Fig**. 13C;
Group 8) except summer. Rest of the stations clustered randomly with each other due to their
different gradients.

**4 Discussions**

**4.1 Seasonal effect on TEP distribution**

Study of seasonal trends on EPS (exopolymeric substances; equivalent to TEP)
confirmed the formation of EPS at earlier season in upper sea column with time (Riedel et al.
2006, Collins et al. 2008). Traditionally, the resource of TEP and their precursors are
phytoplankton cells, especially under bloom situations (Hong et al. 1997; Passow 2002a;
Passow and Alldredge 1994). In BS, *Skeletonema costatum* and *Coscinodiscus oculus-iridis*
during winter. *Noctiluca scintillans, Chaetoceros affinis, Chaetoceros* sp. through all seasons





were reported as dominant phytoplankton species (Yang et al. 2018) which may have local
influence on CTEP (Passow 2002a). However, *P. sulcata* showed dominancy in BS through 3
seasons BS by corresponding closely with TEP (Fig. 6D, 6E & 6F). During autumn at NYS,
*Pseudo-nitzschia pungens* and *Proboscia alata* were reported dominant at the same location
of dense CTEP compared to present study (Li et al. 2017) which may also liable for CTEP
assemblages by demonstration close relation in CCA (Fig. 6G). In SYS, dominancy of
*Paralia sulcate* and *Thalassiosira angulate* (Li et al. 2017, Liu et al. 2015a) with *Pseudo-*
*nitzschia pungen* (Li et al. 2017) were reported at same magnitudes during autumn which
were similar to present study and also maintained close correspondences with TEP (Fig. 6J).
Coastal SYS showed the dominancy of *Skeletonema costatum* and *Thalassiosira*
*nordenskiodii* during winter (Wen et al. 2007). Among phytoplankton, cyanobacteria (36%)
were reported stratified during summer at SYS (Liu et al. 2015b). Phytoplankton i.e. *Paralia*
*sulcata*, *Prorocentrum dentatum* and *Thalassiosira angulata* were abundant species at south
of SYS which location were highly concentrated with TEP (**Fig**. 5A) according to present
study. Biological process of these species may liable for the abundance of TEP along with
those study areas.

However, CTEP can be high in lower biological activity. In absence of

phytoplankton, dissolve organic matter can be source of TEP (Wurl et al. 2011b). During
autumn and summer, CTEP was abundant in this study in where nutrients were higher and
Chl-a was low. Arctic autumn showed low TEP concentrations in upper water layer with no
significant enrichment (Wurl et al. 2011). Though, limited sampling data showed no
significant correlation of TEP with nutrients and salinity in Changjinag (Yangtze River)
estuary (CE) near East China Sea (Guo & Sun 2018). However, CTEP was higher along
CDW from CE during this study at summer. Consumption by various organisms (Tranvik et
al. 1993) can also change TEP distribution. In some places during summer and autumn, due
to low nutrient concentration, organisms (**Fig**3) may feed on TEPs which is why CTEP (**Fig**.
3) is very low in those areas. Higher tempered zone also possessed high TEP production due
to the effect of temperature on photosynthetic parameters (Claquin et al. 2008, Fukao et al.
2012). So, CTEP has been observed high during summer and spring than other seasons in
various estuaries and seas (Table 8). In SYS, CTEP was higher in subsurface area with low
Chl-a (**Fig**. 3E) but high Temperature (**Table 1**) at YSCC and CDW during summer 2015.
However, changes in CTEP through seas result from a balance between sources i.e.
production by algae, bacteria, and possibly other organisms (Ortega-Retuerta et al. 2010)





which supported the present data during winter in study areas. Arctic winter season also
showed the highest water column TEP concentrations and formation rates until spring (Wurl
et al. 2011b). On the other hand, CTEP in spring was lower in Changjinag (Yangtze River)
estuary near East China Sea than in summer (Guo & Sun 2018).
Studies (Table 4) showed that highest CTEP was found at the surface water column in
Adriatic Sea (Radic et al. 2005) and lowest in Weddell Sea (Ortega-Retuerta et al. 2009). In
North Pacific Ocean, surface possessed higher CTEP than below 50 meters (Table 5).
Average CTEP was higher (Table 4) in western subarctic part (Ramaiah et al. 2005) than
eastern (Wurl et al. 2011b) and western tropical parts (Kodama et al. 2014) as well as eastern
subarctic zone (Wurl et al. 2011b). Higher estuarine CTEP was found in Changjinag River
Estuary (Guo & Sun 2018) during both in summer and spring rather than that of Jiulong
River estuary (Peng and Huang 2007) and Pearl River estuary (Sun et al. 2010). In the surface
of Bay areas, present study observed lower average CTEP in Bohai Sea than in Chesapeake
Bay (Malpezzi et al. 2013) and Gulf of Cadiz (Garc et al. 2002). Gulf of Aqaba showed
highest CTEP (Bar-Zeev e.t al. 2009) below 50 meters of any seas (Table 5). Below 100m,
CTEP was higher in Eastern Mediterranean Sea (Bar-Zeev et al. 2011) than other part of this
sea (Ortega-Retuerta et al. 2010) and Gulf of Aqaba (Bar-Zeev et al. 2009). Average vertical
CTEP profiling (0-100m) was higher in Ross Sea (Hong et al. 1997) than rest seas (Table 6).
Rather than in summer 2015 and autumn 2014, Present study observed high CTEP at BSCC
in Bohai Sea as well as at LCC in NYS in winter 2015. Combined effects of seasonal
environmental parameters may cause these variations through those water columns.
**4.2 Seasonal sinking rate variations of TEP**
Sinking rates or particle sedimentation of TEP can also cause changes in TEP
distribution (Passow et al. 2001). SETCOL method was most popular scientific method
(Table 7) to track it (Guo & Sun 2018) due to its simplicity and reliability. However, motion
and turbulence of seawater was ignored in SETCOL which have complex effect on particle
sinking in ocean (Javier et al. 1996, Ruiz et al. 2004). So, the actual situation remained
unclear with theories (Guo & Sun 2018). Seawater is denser than TEP (density 0.70-0.84 g
cm$^{-3}$) which indicated that pure TEP will ascend upward in ballast free condition (Azetsu-
Scott and Passow 2004). So, sinking rates of TEP can be negative (Azetsu-Scott and Passow
2004, Mari 2008). In real scenario, presence of organic and inorganic matter in seawater
make complex situation for TEP to be pure. Sticky gel characteristics of TEP (Engel 2000,





Rochelle-Newall et al. 2010) may aggregated them with various detritus, particles and
organisms i.e. bacteria, phytoplankton and mineral clays (Prieto et al. 2002) which may
influence them to sink downward in water (Mari et al. 2017).

Freshwater lake has lower particle concentration with higher sinking rates of TEP

(Table 7) due to the influence of phytoplankton cells aggregation (Vicente et al. 2009).
Estuarine TEP sedimentation rate was reported lower in spring than other seasons (Guo &
Sun 2018). Average sinking rate of TEP in NYS (Table 9) was observed higher in summer
and winter during present study may be due to higher salinity (**Table 1**) and primary
productivity (Chl-a). In SYS, highest TEP sedimentation was at Station H38 (2.40 mD$^{-1}$) may
be caused by counter effect of CDW and CWC (**Fig**. 2). SYS also has higher sinking rates of
TEP in autumn than BS and NYS may be due to high nutrient concentrations (**Fig**. 4) at the
bottom that may stick with TEP to sink downwards. Rather than other coastal water
(Changjiang Estuary), Bohai Sea possessed higher sinking rates of TEP in average, especially
during summer (Table 9). Seasonal effect on concentrations of environmental parameters and
coastal currents' activity may cause these differences in sedimentation rates of TEP in study
areas.

## 4.3 Potential role of seasonal carbon export associated with TEP

Organic carbon formation in sea surface associated with TEP and its sedimentation is

a complex part of carbon cycle in ocean (Mari et al. 2017).  These exopolymer particles
contained carbon complex which may disappeared due to TEP sedimentation and degradation
by bacteria (Prieto et al. 2006) in euphotic zone. Due to alignment of TEP as same magnitude
of phytoplankton cells abundance (Passow 2002b, Passow et al. 2001), TEP sinking was
accepted as important carbon pathway for its dominant effect on TEP-C (Stoderegger and
Herndl 1999, Obernosterer and Herndl 1995, Guo & Sun 2018). Stickiness of TEP and its
balances between production and degradation rates contributed in POC cycling (Mari et al.
2017) in the ocean. TEP sedimentation roughly contributes 30% in POC flux at Santa
Barbara Channel (Passow et al. 2001) and 0.02%-31% in oligotrophic reservoir of southern
Spain (Mari et al. 2017). TEP-C was lower in spring than in summer at Changjiang River
(Yangtze River) estuary near East China Sea. Present study observed higher total average of
TEP-C in NYS than BS and SYS especially in winter and lowest TEP-C in autumn in BS
during all three seasons. Considering the complex effect of all environmental parameters,
TEP-C distribution showed similar correlations with nutrients and Chl-a as CTEP in different




seasons accordingly. Results of present study suggest the importance of TEP in POC cycle in
Bohai Sea and Yellow Sea compared to phytoplankton cells and zooplankton fecal pellets
(Turner 2002, Turner 2015). TEP controlled the biological carbon pump of atmospheric $CO_2$
(Mari et al. 2017). With significant seasonal TEP-C distribution, the data showed an
unavoidable importance of TEP and its sedimentation rates for exporting carbon in study
areas.
**5 Conclusions**
Seasonal variations of TEP concentration was complex and mostly depends on
nutrients and Chl-a. Correlations on the basis of 168 samples of TEP with same amount of
other environmental parameters showed variations among seas as well as seasons.
Temperature varied from 0-28 ºC round the year but TEP stacked from 0 to below 10 μg Xeq.
$L^{-1}$ in average. Chl-a may liable for TEP distribution during autumn and summer, especially
in SYS and nutrients to TEP in winter in BS. Coastal current mixing has an influence on
CTEP due to its dominancy at dilution zones. Sinking rates of TEP mostly varied at surface
of BS and NYS during summer and winter. SYS has moderated sinking rates of TEP at its
surface and bottom with higher nutrients concentrations. With close correspondences,
dominant phytoplankton i.e. *P. sulcata, T. excentrica* and *R. styliformis* have influences of
high TEP concentration. Average carbon exports maintained same magnitudes with TEP
during each season. Average TEP sinking was diverse at SCM but higher at surface through
all seasons. Further research on POC cycle by measuring CTEP, TEP-C and its sinking rates
with seawater density and turbidity of selected study area are recommended to be more
precise on carbon contributions of exopolymers in the process of biological and chemical
carbon pump in open and coastal seas.
**Acknowledgements**
This study was supported by National Natural Science Foundation of China (Nos.
41876134, 41676112 and 41276124), the Key Project of Natural Science Foundation for
Tianjin (No. 17JCZDJC40000), the University Innovation Team Training Program for
Tianjin (TD12-5003) and the Tianjin 131 Innovation Team Program (20180314), and the
Changjiang Scholar Program of Chinese Ministry of Education to Jun Sun. The authors are
grateful to Prof. Houjie Wang for providing the data of temperature and salinity, and also
thank the crew and captain of the R/V Dongfanghong 2 for their assistance in sample




collection during the cruise supported by National Natural Science Foundation of China
(Nos. NORC2014 and NORC2015).

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




**Table 1** Average concentration of environmental parameters along with related seas during study seasons.

| Seasons | Seas | Temperature (°C) | Salinity (PSU) | DIP (μmol/l) | DIN (μmol/l) | DSi (μmol/l) | Ammonium (μmol/l) | Nitrite (μmol/l) | Nitrate (μmol/l) | Chl-a (μg/l) |
|---|---|---|---|---|---|---|---|---|---|---|
| Autumn (2014) | Bohai Sea | 9.77 | 24.13 | 0.29 | 10.24 | 7.24 | 0.98 | 0.18 | 9.08 | 0.34 |
| | North Yellow Sea | 13.23 | 31.56 | 0.18 | 3.91 | 3.70 | 0.45 | 0.27 | 3.19 | 0.35 |
| | South Yellow Sea | 16.32 | 31.60 | 0.22 | 4.95 | 5.64 | 0.58 | 0.16 | 4.21 | 0.78 |
| Summer (2015) | Bohai Sea | 23.90 | 30.55 | 0.08 | 3.94 | 3.91 | 0.85 | 0.63 | 2.45 | 1.02 |
| | North Yellow Sea | 20.40 | 31.71 | 0.14 | 2.01 | 2.55 | 0.53 | 0.27 | 1.21 | 0.83 |
| | South Yellow Sea | 21.40 | 31.28 | 0.19 | 4.15 | 4.01 | 0.57 | 0.23 | 3.35 | 1.42 |
| Winter (2015) | Bohai Sea | 0.29 | 6.43 | 0.29 | 4.28 | 4.28 | 1.78 | 0.19 | 2.31 | 4.82 |
| | North Yellow Sea | 4.73 | 32.20 | 0.14 | 1.06 | 3.50 | 0.47 | 0.09 | 0.49 | 7.00 |



**Table 2** Variations of average TEP concentrations and its sinking rates in different seas during study.

| Seasons | | Ranges | BS | NYS | SYS |
|---|---|---|---|---|---|
| **CTEP (μg Xeq. L⁻¹)** | | Average | 2.08 | 3.32 | 1.18 |
| | | Maximum | 14.75 | 23.20 | 11.99 |
| | | Minimum | 0.20 | 0.20 | 0.19 |
| **Sinking rate of TEP (m d⁻¹)** | | Average | 0.98 | 1.03 | 1.09 |
| | | Maximum | 2.06 | 2.47 | 2.40 |
| | | Minimum | 0.64 | 0.72 | 0.62 |



**Table 3** Variations of average TEP concentrations and its sinking rates in different seasons.

| Seasons | | Ranges | Sinking rate of TEP (m d⁻¹) | CTEP (μg Xeq. L⁻¹) |
|---|---|---|---|---|
| Total | Average | Average | 1.03 | 2.13 |
| | | Maximum | 2.47 | 23.20 |
| | | Minimum | 0.62 | 0.20 |
| Autumn | 2014 | Average | 1.02 | 0.67 |
| | | Maximum | 1.55 | 11.99 |
| | | Minimum | 0.72 | 0.00 |
| Summer | 2015 | Average | 1.04 | 1.10 |
| | | Maximum | 2.47 | 12.39 |
| | | Minimum | 0.62 | 0.00 |
| Winter | 2015 | Average | 1.03 | 6.17 |
| | | Maximum | 2.06 | 23.20 |
| | | Minimum | 0.64 | 0.20 |





**Table 4.** Concentration of TEP in 0-50m depths at different area from various reports.

| Location | CTEP (µg Xeq. L$^{-1}$) | References |
|---|---|---|
| Santa Barbara Low Strait | 85-252 | Passow and Alldredge (1995) |
| The Baltic Sea | 145-322 | Engel (2002) |
| Gulf of Cadiz | 25-717 | Garc et al. (2002) |
| Eastern North Atlantic | 20-60 | Engel (2004) |
| Adriatic Sea | 4-14800 | Radic et al. (2005) |
| Western subarctic North Pacific | 40-190 | Ramaiah et al. (2005) |
| Arabian Sea | 507-560 | Prieto et al. (2006) |
| Jiulong river estuary | 530-720 | Peng and Huang (2007) |
| Weddell Sea | 0-48.9 | Ortega-Retuerta et al. (2009) |
| Newson Estuary | 805-1801 | Wetz et al. (2009) |
| Gulf of Aqaba | 130-222 | Bar-Zeev et al. (2009) |
| Pearl River Estuary | 85-1235 | Sun et al. (2010) |
| Mediterranean Sea | 19.4-53.1 | Ortega-Retuerta et al. (2010) |
| Eastern tropical North Pacific | 22.5 | Wurl et al. (2011b) |
| Eastern Mediterranean Sea | 116-420 | Bar-Zeev et al. (2011) |
| Eastern subarctic North Pacific | 28.7 | Wurl et al. (2011b) |
| Chesapeake Bay | 37-2820 | Malpezzi et al. (2013) |
| Western tropical North Pacific | 43.3 | Kodama et al. (2014) |
| Changjiang Estuary | 173.33-1423.33 | Guo & Sun (2018) |
| Bohai Sea | 0.19-14.75 | This Study (2014-15) |
| North Yellow Sea | 0.19-23.20 | This Study (2014-15) |
| South Yellow Sea | 0.19-11.99 | This Study (2014-15) |





**Table 5.** Concentration of TEP in 50-100 depths at different area from various reports.

| Location | CTEP ( µgXeq. L$^{-1}$) | References |
|---|---|---|
| Gulf of Aqaba | 106-228 | Bar-Zeev e.t al. (2009) |
| Mediterranean Sea | 9.1-94.3 | Ortega-Retuerta et al. (2010) |
| Eastern tropical North Pacific | 9.2 | Wurl et al. (2011b) |
| Eastern Mediterranean Sea | 48-189 | Bar-Zeev et al. (2011) |
| Eastern subarctic North Pacific | 11.6 | Wurl et al. (2011b) |
| Western tropical North Pacific | 42.2 | Kodama et al. (2014) |
| Bohai Sea | 0.39-4.33 | This Study (2014-15) |
| North Yellow Sea | 0.19-11.01 | This Study (2014-15) |
| South Yellow Sea | 0.2-0.79 | This Study (2014-15) |





**Table 6.** Average concentrations of TEP at different depths in different area.

| Location | Water layer (m) | CTEP ($\mu g\ Xeq.\ L^{-1}$) | References |
|---|---|---|---|
| Santa Barbara Low Strait | 0-75 | 29-68 | Passow andAlldredge (1995) |
| Ross Sea | 0-150 | 1003-7667 | Hong et al. (1997) |
| Northwest Atlantic | 10--70 | 10-120 | Engle (2004) |
| Bransfield Strait | 0-100 | 0-346 | Corzo et al. (2005) |
| Gulf of Aqaba | 100< | 23-209 | Bar-Zeev et al. (2009) |
| Mediterranean Sea | 100< | 4.5-23.5 | Ortega-Retuerta et al. (2010) |
| Eastern Mediterranean Sea | 100< | 83-386 | Bar-Zeev et al. (2011) |
| North Bering Sea | - | 34-628 | Lili et al. (2012) |
| Changjiang Estuary | 0-100 | 321.52 | Guo & Sun 2018 |
| Bohai Sea | 1-100 | 0.2-14.75 | This Study (2014-15) |
| North Yellow Sea | 1-100 | 0.2-23.20 | This Study (2014-15) |
| South Yellow Sea | 1-100 | 0.19-11.99 | This Study (2014-15) |





**Table 7.** Variations of Sinking rates of TEP and related applied methods from different reports.

| Location | Sinking rate of TEP (m d$^{-1}$) | Method | References |
|---|---|---|---|
| Santa Barbara Strait | -0.22-0.04 | SETCOL | Azetsu-Scott 2004 |
| South Pacific Ocean | -0.29~0.49 | SETCOL | Mari 2008 |
| Freshwater Lake Quéntar | 1.12~1.31 | Sediment Trap | Vicente 2009 |
| Changjiang Estuary | 0.08-1.08 | SETCOL | Guo & Sun 2018 |
| Bohai Sea | 0.64~2.06 | SETCOL | This Study (2014-15) |
| North Yellow Sea | 0.72~2.47 | SETCOL | This Study (2014-15) |
| South Yellow Sea | 0.62~2.40 | SETCOL | This Study (2014-15) |




**Table 8.** Seasonal average TEP distribution in surface area from various research with their minimum to maximum ranges.

| Seasons | Location | CTEP (µg Xeq. L$^{-1}$) | | | References |
|---|---|---|---|---|---|
| | | Minimum | Maximum | Average | |
| Autumn | Bohai Sea | 0.00 | 0.98 | 0.42 | This Study (2014-15) |
| | North Yellow Sea | 0.00 | 2.75 | 0.44 | This Study (2014-15) |
| | South Yellow Sea | 0.00 | 11.99 | 0.93 | This Study (2014-15) |
| Spring | Changjiang Estuary | 173.33 | 840.00 | 506.67 | Guo & Sun 2018 |
| | Northeast coast of Japan | 901.00 | 1142.00 | 1021.50 | Ramaiah et al. 2001 |
| | Southern Iberian coasts | 507.00 | 560.00 | 533.50 | Prieto et al. 2006 |
| Summer | Changjiang Estuary | 473.33 | 1423.33 | 948.33 | Guo & Sun 2018 |
| | Baltic Sea | 145.00 | 322.00 | 233.50 | Engel 2002 |
| | Northeast Atlantic Ocean | 10.00 | 120.00 | 65.00 | Engel 2004 |
| | Bransfield Strait | 0.00 | 346.00 | 173.00 | Corzo et al. 2005 |
| | Jiulong estuary | 530.00 | 720.00 | 625.00 | Peng and Huang 2007 |
| | Pearl River estuary | 85.00 | 1235.00 | 660.00 | Sun et al. 2010 |
| | Bohai Sea | 0.00 | 12.39 | 1.02 | This Study (2014-15) |
| | North Yellow Sea | 0.00 | 6.49 | 0.83 | This Study (2014-15) |
| | South Yellow Sea | 0.00 | 10.22 | 1.42 | This Study (2014-15) |
| Winter | Bohai Sea | 0.20 | 14.75 | 4.82 | This Study (2014-15) |
| | North Yellow Sea | 0.39 | 23.20 | 7.00 | This Study (2014-15) |



**Table 9.** Seasonal average sedimentation rates of TEP in different locations with their maximum and minimum ranges from different reports.

| Seasons | Location | TEP sinking rates (mD$^{-1}$) | | | References |
|---------|----------|---------|---------|---------|------------|
| | | **Minimum** | **Maximum** | **Average** | |
| Autumn | Bohai Sea | 0.72 | 1.16 | 0.90 | This Study (2014-15) |
| | North Yellow Sea | 0.76 | 1.19 | 0.95 | This Study (2014-15) |
| | South Yellow Sea | 0.78 | 1.55 | 1.13 | This Study (2014-15) |
| Spring | Changjiang Estuary | 0.08 | 0.57 | 0.33 | Guo & Sun 2018 |
| Summer | Changjiang Estuary | 0.10 | 1.08 | 0.59 | Guo & Sun 2018 |
| | Bohai Sea | 0.86 | 1.57 | 1.01 | This Study (2014-15) |
| | North Yellow Sea | 0.77 | 2.47 | 1.06 | This Study (2014-15) |
| | South Yellow Sea | 0.62 | 2.4 | 1.05 | This Study (2014-15) |
| Winter | Bohai Sea | 0.64 | 2.06 | 1.02 | This Study (2014-15) |
| | North Yellow Sea | 0.72 | 1.53 | 1.03 | This Study (2014-15) |





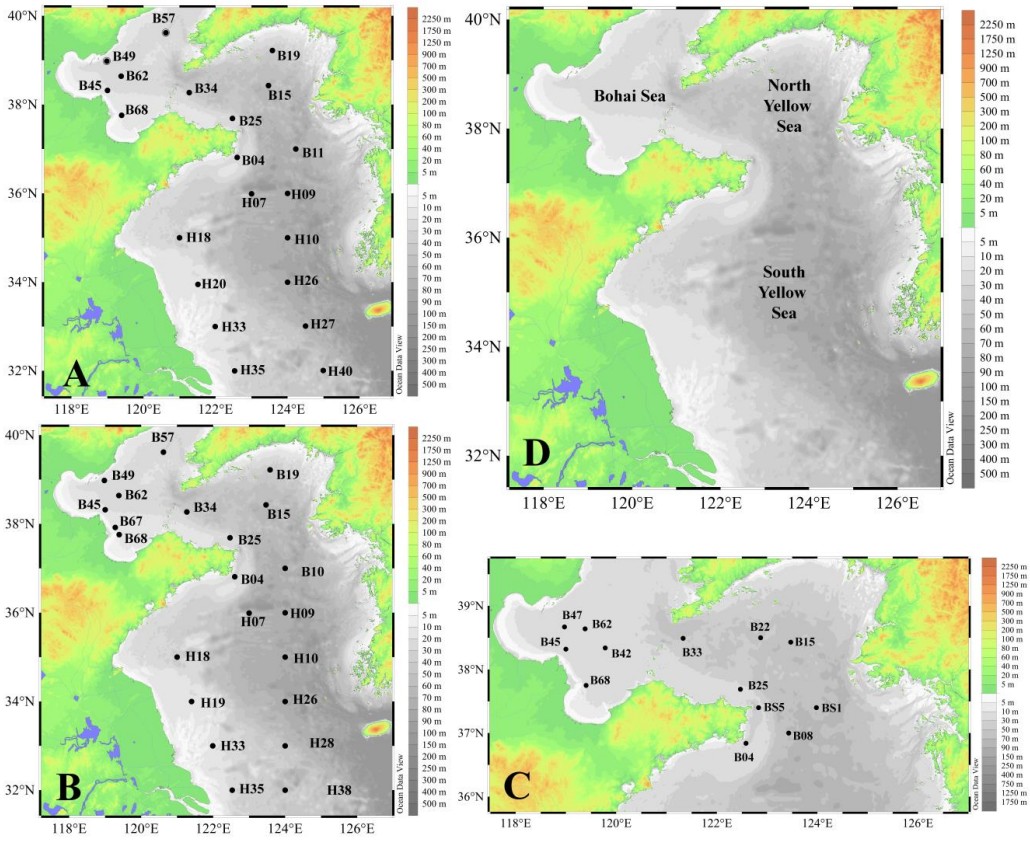

**Figure 1.** Study stations in map of autumn 2014 (A), summer 2015 (B) and winter 2015 (C) with their oceanic segmentation (D).





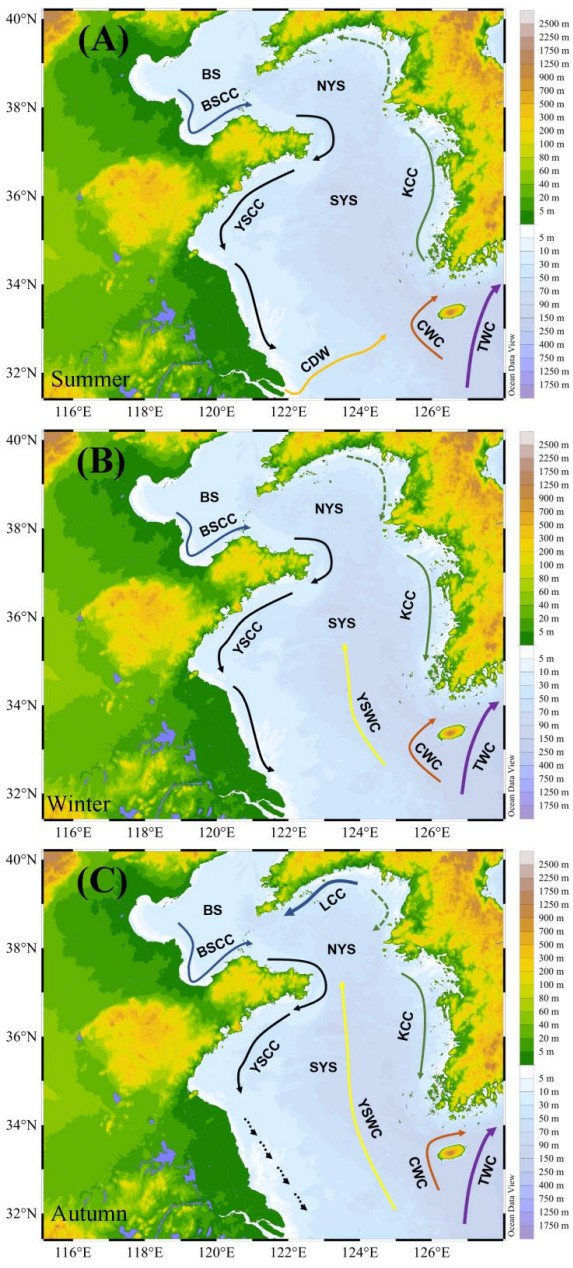

**Figure 2.** Direction of Currents at Bohai Sea (BS=Bohai Sea, BSCC=Bohai Sea coastal current), North Yellow Sea (NYS=North Yellow Sea, KCC=Korean Costal Current, LCC=Liaonan coastal current, YSCC=Yellow Sea Costal Current) and South Yellow Sea (SYS=South Yellow Sea, YSWC=Yellow Sea Warm Sea, CDW=Changiang Diluted Water,





CWC= Cheju Warm Current, TWC=Tsushima Warm Current) during summer (a), winter (b) and autumn (c) seasons (Collaboratively modified after Hwang et al. 2014; Su 1998; Yuan et al. 2008; Isobe 2008, Zhang et al. 2003).

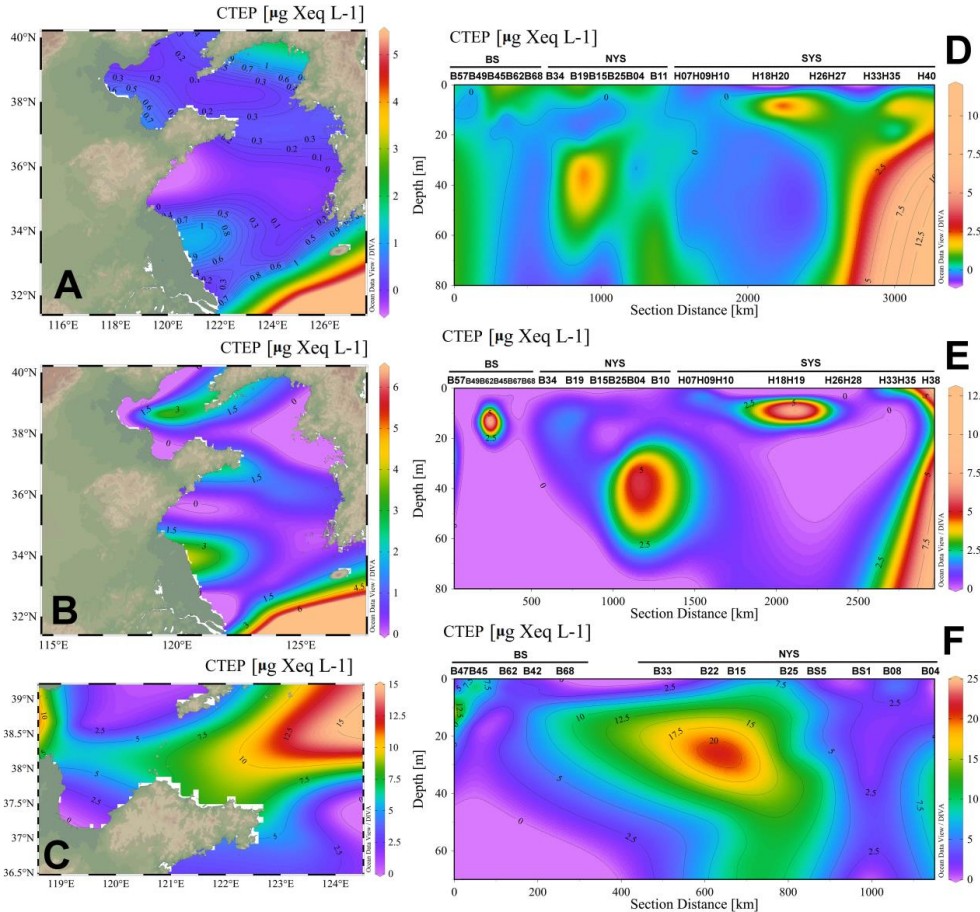

**Figure 3.** Average seasonal concentrations of CTEP (A, B & C) with sectional view (D, E & F) of BS (D), NYS (E) and SYS (F) during autumn (A), summer (B) and winter (C).



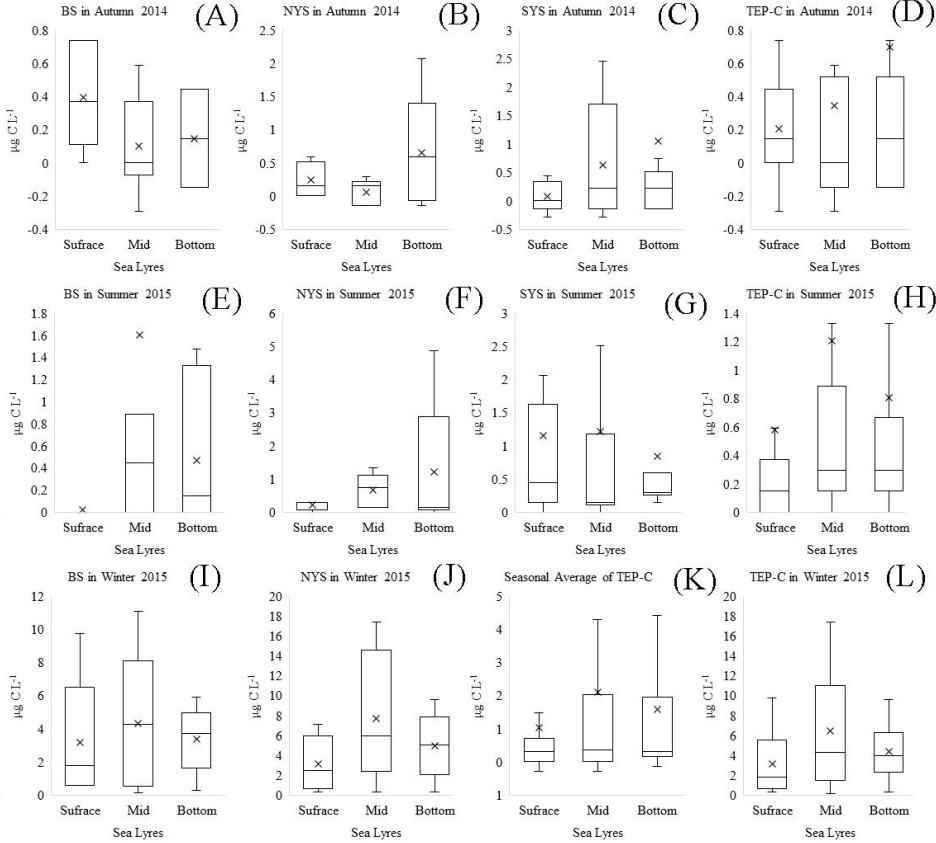

**Figure 4.** TEP associated carbon concentration during all seasons in Bohai Sea (A=autumn 2014, E=summer 2015, I=winter 2015), North Yellow Sea (B=autumn 2014, F=summer 2015, J=winter 2015), South Yellow Sea (C=Autumn 2014, G=summer 2015) with all data in average (K), autumn 2014 (D), summer 2015 (H) and winter 2015(L).





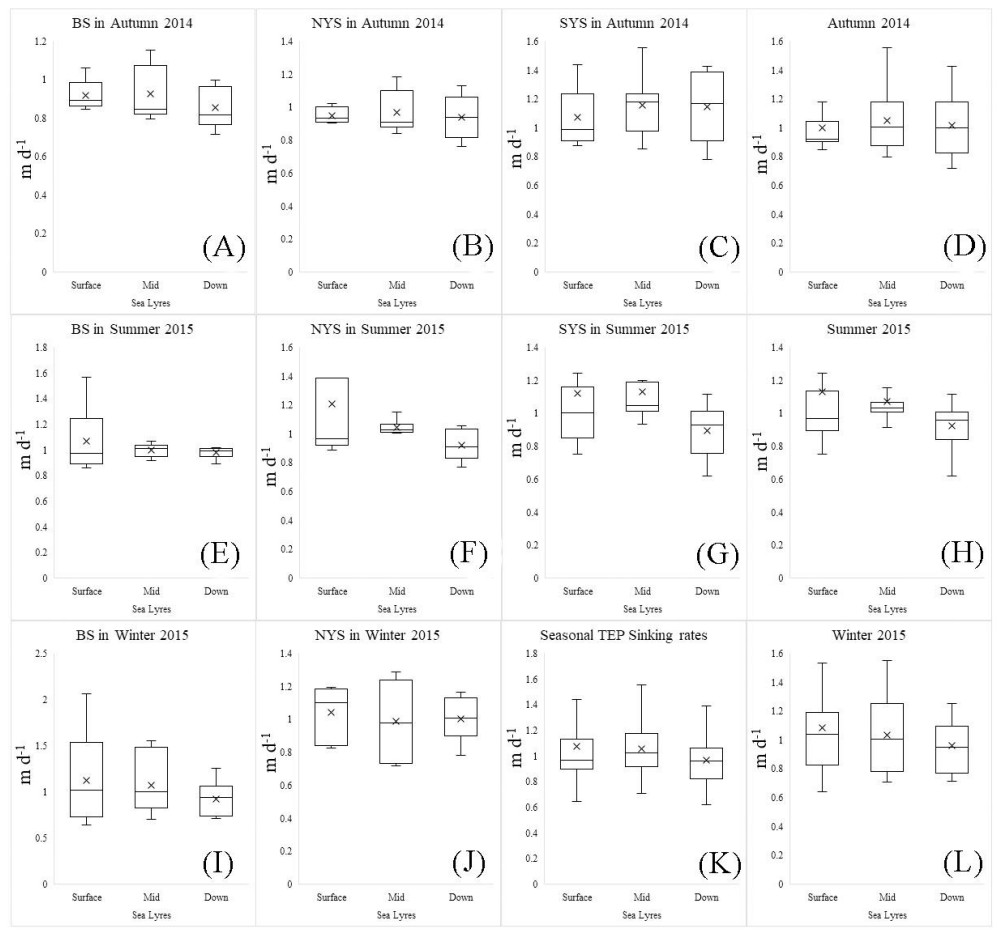

**Figure 5.** TEP sinking flux during all seasons in Bohai Sea (A=autumn 2014, E=summer 2015, I=winter 2015), North Yellow Sea (B=autumn 2014, F=summer 2015, J=winter 2015), South Yellow Sea (C=autumn 2014, G=summer 2015) with all seasonal sinking data in average (K), autumn 2014 (D), summer 2015 (H) and winter 2015(L).



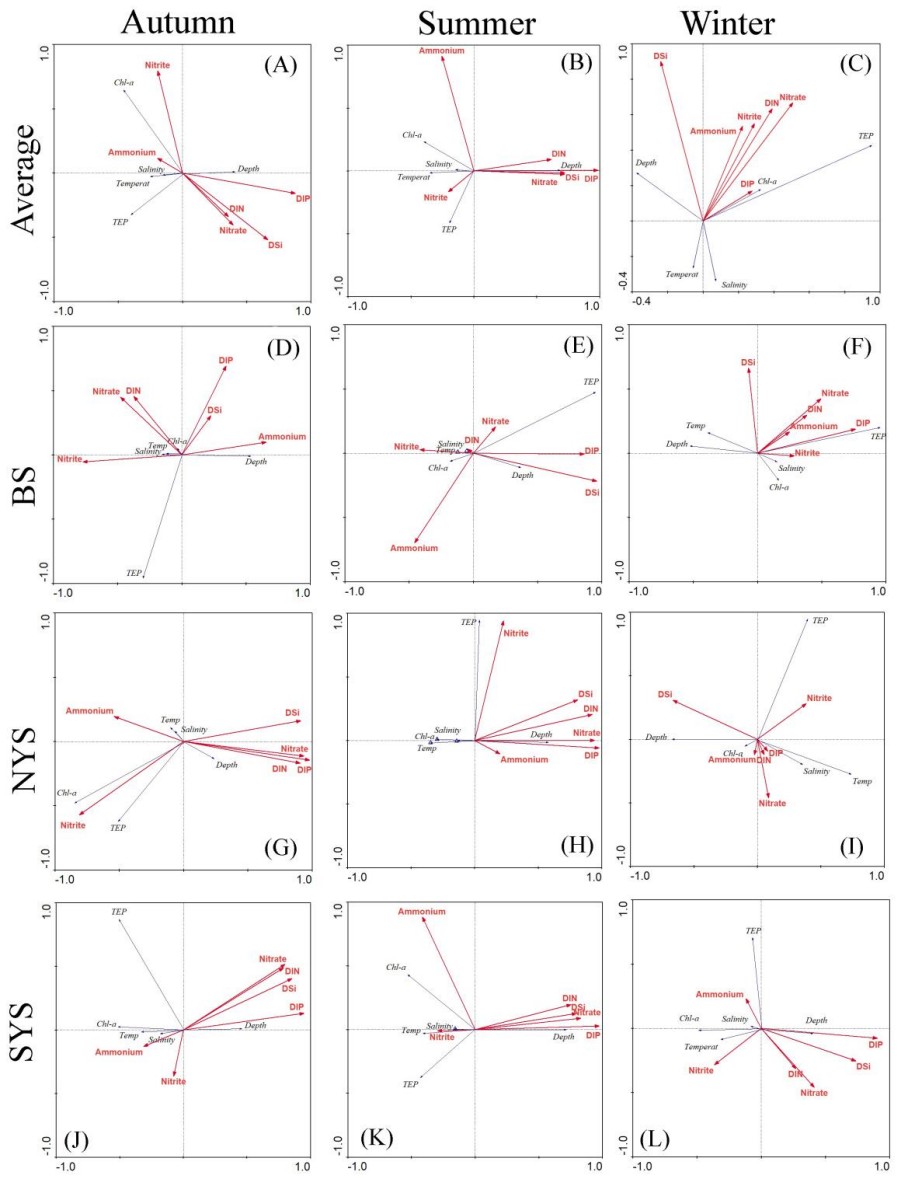

**Figure 6.** CCA analysis of all parameters of all study stations. Seasonal average i.e. autumn 2014 (A), summer 2015 (B), winter 2015(C) and total (L) with the CCA of all parameters according to locations i.e. Bohai Sea (D=autumn 2014, E=summer 2015, F=winter 2015), North Yellow Sea (G=Autumn 2014, H=summer 2015, I=winter 2015) and South Yellow Sea (J=autumn 2014, K=summer 2015).





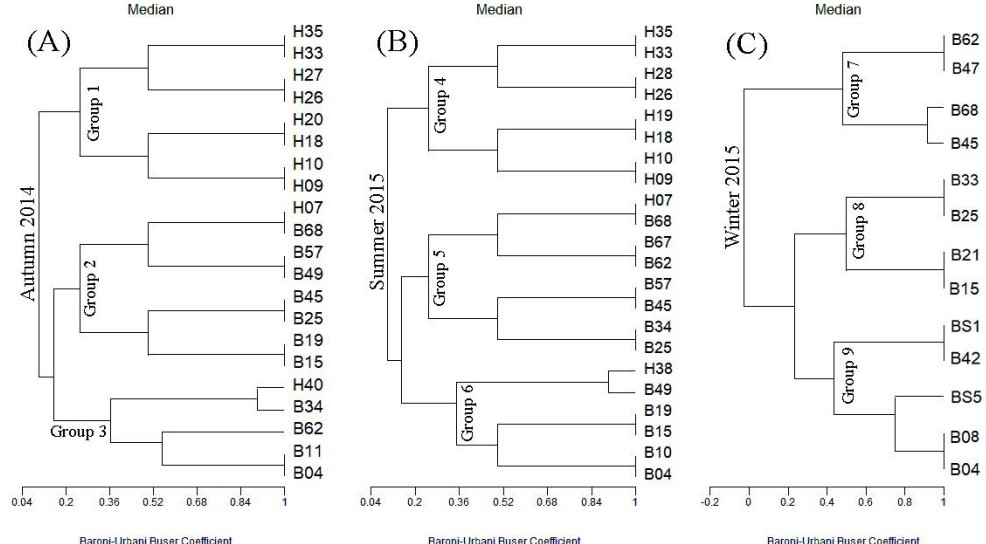

**Figure 7.** Seasonal Cluster analysis among related study stations of study area by considering all parameters and CTEP, segmented by seasons i.e. autumn 2014 (A), summer 2015 (B) and winter 2015 (C). Groups (1-9) are indicating clustered group of closely related stations.