# Peer review of "Seasonal Sinking rates of Transparent Exopolymer Particles (TEP) concentrations with associated Carbon flux in adjacent Bohai Sea and Yellow Sea"

_Biogeosciences, 2019_

## Referee Comment (RC1) · Anonymous Referee #1 · 5 Apr 2019

This is an interesting study on the seasonal distributions of transparent exopolymer particles and its impacts on carbon cycle in the north Chinese seas. The quality of the dataset, although lacking in my opinion of important aspects needed to address TEP distribution driving factors in the study area, is good and nicely approached to answer the specific scientific questions raised by the study. However, the manuscript has major drawbacks regarding its overall structure and results interpretation and I suggest that it could be considered for final publication in BG journal, but following major revision. Most importantly, the English is very poorly phrased throughout the text, making the

manuscript hard to follow for the reader in many cases. I started to suggest a few sentence rewrites, but there are many others that need rewriting. I strongly suggest having a native English speaker do a major edit on the final version of the manuscript before resubmission.

Please follow my major and other comments below.

INTRODUCTION SECTION

In the introduction part a large discussion is introduced on the SML. However the SML is beyond the scopus of this study. I suggest that the simple reference introduced in lines 46-47 is fairly enough on this and all other related text is removed from the manuscript (lines 64-75 and elsewhere).

- Line 38: between the - Line 38: organic carbon pools - Line 41: considered - Line 43: sourced? Please rephrase - Line 47: organically? Please rephrase - Line 51: control - Line 54: Please rephrase - Line 60: Please rephrase - Line 62: at the sea surface - Line 63: Mari et al., 2017 review addresses this issue and should be added in the references herein - Line 78: in the water column - Line 84: at the upper water column - Line 85: in summer rather than spring - Line 86: on carbon export - Line 87: The present study - Line 92: East of the BS - Lines 95-96: Please rephrase - Line 104: on TEP - Lines 108-110: Correct, but poor correlations amongst TEP and other environmental parameters has also be attributed to other biotic factors driving their distributions such as microbial breakdown of larger marine snow particles, sloppy feeding by meso- and macrozooplankton, abiotic TEP formation by bacteria and also consuming of TEP by bacteria. Please see review article of Mari et al., 2017 and references therein as an example for the relevant discussion. I suggest that a relevant part is being introduced in the introduction of the manuscript instead of the extended discussion on SML.

MATERIALS AND METHODS SECTION

- It is unusual in a manuscript not to have a Table outline of the sampled stations, coordinates and corresponding sampling depths. Of course this will be too long for the main body of the text and I suggest that the authors should considering adding this information as supplementary material. - In my opinion the authors should consider merging sections 2.1 and 2.2 in one section outlining study area and conducted samplings. - Lines 122-127: This part should be moved to the results section. Please follow my other comment on this issue below. - Line 141: nutrients separately? Please rephrase - Line 142: sampling bottles - Line 145: Please correct nomenclature to mL here and throughout the text. Also use L (capital) for litres throughout the text and figures/tables. - Lines 149-152: Please rephrase. The same text is introduced twice and a part of it is also being repeated above in lines 143-144 regarding Chl-a - Lines 159-160: dissolved, in all cases - Lines 162-164: Please rephrase - Line 169: were then soaked - Line 170: H2SO4 for sulphuric acid - Line 170: were gently - Line 171: lied - Lines 183-184: Please rephrase - Lines 195-201: I suggest that the authors should consider removing this text from this part. - Section 2.6: I suggest that the authors should consider moving the TEP-C calculations part above in sect. 2.4 and the phytoplankton species description part in the relevant sect 2.3 above.

RESULTS SECTION

An interesting aspect of this study in my opinion is the highlighted connection between the TEP distributions patterns and water masses characteristics in some cases. I suggest that the authors should consider elaborating a little bit more on this issue maybe providing a figure highlighting this connection. Please see the study of Parinos et al., 2017. Cont Shelf Res., 149, 112-123 for an approach on this matter. Please also check if this connection is highlighted/displayed on the clustering of stations in figure 7.

- Section 3.3: I suggest that the authors should consider merging this section with the previous one, sect 3.2, since it is expected that TEP-C distributions will be the same ones as in the case of CTEP since TEP-C=CTEPx0.75 in all cases

- Section 3.6: please check corresponding figure numbers! Fig. 16 is Fig. 6?? Fig. 13 is Fig. 7? Also please follow my comment below ''section 4.1" regarding lines 322-327

DISCUSSION SECTION

- Line 337: EPS include also protein-containing Coomassie stainable particles so the term EPS in not equivalent to TEPs only. TEPs are a part of EPS. - Line 339: resource? Please rephrase

-Section 4.1: In this section, see also my comment above on lines 322-327, I cannot see the connection between the various phytoplankton species and TEP distributions patterns on Fig.6. Therein the CCA analysis is nicely presented but focused on TEP-Chla relations and not individual species. I suggest that the authors should consider adding a table outlining TEP-phytoplankton species correlations in support to the discussion introduced in this section or re-phrase the paragraph.

- Line 345: the linkage between TEP and Chl-a in Figs. 6d-3-f is not at all strong in my opinion. - Line 348: Same as above for figure 6g. - Lines 374-376: but also, other than sources, to other factors consuming TEPS, i.e. consuming of TEP by bacteria, which is an aspect that cannot be addressed based on the presented dataset.

- Section 4.3: My feeling is that in order to address the potential role of TEP-C in the overall carbon cycle in the study area an essential aspect that is missing from the dataset is considering TEP carbon in respect to POC values. Taking into account the low TEP concentrations recorded overall in the study area, up to 23.2 $\mu$g XG eq L-1, and the high chl-a values recorded in summer and moreover the very high chl-a values recorded during winter 2015, TEP-C seems that it probably contributes a very small fraction to POC. Please see Ortega-Retuerta et al., 2010;2019, Bar-Zeev et al., 2011 and references therein, amongst others, as an example for the relevant discussion. If there are POC data available for the studied samples then they should be in my opinion included in the dataset and discussed in the manuscript. Overall, I believe that the relevant discussion/ interpretations introduced in section 4.3 are not fully supported by

the presented results other than the similarity of TEP distributions with chl-a or nutrients in some cases, which cannot fully support a discussion on TEP role in carbon cycling in the study area.

- Line 426: of the carbon cycle - Line 428: the euphotic zone - Lines 441-442: zooplankton fecal pellets? This statement is not supported by the presented results

TABLES

Nine tables seems a lot for a manuscript. I suggest that the authors should consider some structural changes as i.e.: - merging tables 2 and 3 in one major table presenting the concentrations and sinking rates of TEP reported in this study - merging tables 4-5-6 and 8 in one shorter major table presenting the concentrations of TEPs available from the literature, considering only the relevant data discussed in the text (lines 380-396) - incorporating the information provided in tables 7 and 9, especially table 9 were only data from Guo and Sun other than the ones reported in this study are provided, in the main body of the text (sect 4.2)

FIGURES

Figure 3: Scales used for TEP concentrations should be uniform in all cases, both in min-max values and also scale stepping (step of color change).

END OF REVIEW

---

## Author Comment (AC1) · 23 Apr 2019

**Responses to Reviewer 1**

Authors are like to responses with cordial thanks for these details' scientific comments of reviewer 1. After considering all comments, authors changed the structure of whole manuscript for pursuing a constructive discussion about the main focus of this manuscript. Here are the responses to all creative comments accordingly.

**INTRODUCTION SECTION**

In the introduction part a large discussion is introduced on the SML. However, the SML is beyond the Scopes of this study. I suggest that the simple reference introduced in lines 46-47 is fairly enough on this and all other related text is removed from the manuscript (lines 64-75 and elsewhere).

Answer: Authors rephrased lines 46-477 to 46-49 with necessary references and removed all lines associated with SML. SML only exists at result section 3.2, at line 289 for describing result.

- Line 38: between the

Answer: Changed in line 37.

- Line 38: organic carbon pools

Answer: Changed in line 37.

- Line 41: considered

Answer: Changed in line 40.

- Line 43: sourced? Please rephrase

Answer: Rephrased to "produced" at line 42.

- Line 47: organically? Please rephrase

Answer: Rephrased to "which can make organic sea surface microlayer" at line 46.

- Line 51: control

Answer: Changed in line 52.

- Line 54: Please rephrase

Answer: Rephrased by breaking into 2 lines from 52-56.

- Line 60: Please rephrase

Answer: Rephrased from 60-62.

- Line 62: at the sea surface

Answer: Changed in line 64.

- Line 63: Mari et al., 2017 review addresses this issue and should be added in the references herein

Answer: Authors sited in "Mari et al., 2017" lines 64-65.

- Line 78: in the water column

Answer: Changed in line 68.

- Line 84: at the upper water column

Answer: Changed in line 74.

- Line 85: in summer rather than spring

Answer: Changed in line 75.

- Line 86: on carbon export

Answer: Changed in line 76.

- Line 87: The present study

Answer: Changed in line 77.

- Line 92: East of the BS

Answer: Changed in line 82 by removing it while English language editing.

- Lines 95-96: Please rephrase

Answer: Rephrased to "Zonation of Yellow Sea i.e. North Yellow Sea (NYS) and South Yellow Sea (SYS); will help in visualization of better seasonal scenario." in lines 85-86.

- Line 104: on TEP

Answer: Changed in line 93.

- Lines 108-110: Correct, but poor correlations amongst TEP and other environmental parameters has also be attributed to other biotic factors driving their distributions such as microbial breakdown of larger marine snow particles, sloppy feeding by meso- and macrozooplankton, abiotic TEP formation by bacteria and also consuming of TEP by bacteria. Please see review article of Mari et al., 2017 and references therein as an example for the relevant discussion.

Answer: Authors considered reviewer's advice and added notable lines therein from Mari et al., 2017 with necessary references from lines 95 to 98 i.e. "microbial breakdown (Mari et al. 2017) and phytoplankton species composition (Passow 2002b). Changes in TEP assemblage may also influenced by bacterial secretions and feeding as well as zooplankton grazing locally (Passow and Alldredge, 1999, Surosz et al. 2006, Mari et al. 2017).".

I suggest that a relevant part is being introduced in the introduction of the manuscript instead of the extended discussion on SML.

Answer: Authors removed all lines associated with SML. SML only exists at result section 3.2, at line 289 for describing result.

**MATERIALS AND METHODS SECTION**

- It is unusual in a manuscript not to have a Table outline of the sampled stations, coordinates and corresponding sampling depths. Of course, this will be too long for the main body of the text and I suggest that the authors should considering adding this information as supplementary material.

Answer: Authors added suplimentary data as PDF format with necessary information according to reviewer's requirements.

- In my opinion the authors should consider merging sections 2.1 and 2.2 in one section outlining study area and conducted samplings.

Answer: Authors merged these sections; starting from line 120.

- Lines 122-127: This part should be moved to the results section. Please follow my other comment on this issue below.

Answer: Authors removed this part and added it in result section in short form at lines 197-204.

- Line 141: nutrients separately? Please rephrase

Answer: Rephrased and rearranged at line 123.

- Line 142: sampling bottles

Answer: Changed in line 125.

- Line 145: Please correct nomenclature to mL here and throughout the text. Also use L (capital) for litres throughout the text and figures/tables.

Answer: Rectified through whole manuscript at lines 126, 137, 149 and 154.

- Lines 149-152: Please rephrase. The same text is introduced twice and a part of it is also being repeated above in lines 143-144 regarding Chl-a

Answer: Authors removed previous lines 149-152 and regenerated in new lines 149-151.

- Lines 159-160: dissolved, in all cases

Answer: Changed in lines 144-145.

- Lines 162-164: Please rephrase

Answer: Rephrased in lines 147-149 as "Sextuplicate measurements of TEP were done by following colorimetric method of Passow and Alldredge (1995) for all samples after confirming calibration factor (fx) from xanthan gum curve.".

- Line 169: were then soaked

Answer: Changed and rephrased in line 154

- Line 170: $H_2SO_4$ for sulphuric acid

Answer: Changed in line 155

- Line 170: were gently

Answer: Changed in line 155.

- Line 171: lied

Answer: Changed in line 156.

- Lines 183-184: Please rephrase

Answer: Rephrased and rearranged in lines 170—172 as "The Plexiglass columns were kept undisturbed on the vessel to settle for 2-3 hours. Temperature was maintained through a thermostatically controlled water bath with water jackets by pumping its water.".

- Lines 195-201: I suggest that the authors should consider removing this text from this part.

Answer: Authors removed it to result section 3.1 at line 197-198.

- Section 2.6: I suggest that the authors should consider moving the TEP-C calculations part above in sect. 2.4 and the phytoplankton species description part in the relevant sect 2.3 above

Answer: Thank you for your constructive suggestion. Authors removed lines 202-205 to 161-164 at section 2.3 as advised.

**RESULTS SECTION**

An interesting aspect of this study in my opinion is the highlighted connection between the TEP distributions patterns and water masses characteristics in some cases. I suggest that the authors should consider elaborating a little bit more on this issue maybe providing a figure highlighting this connection.

Answer: Authors considered your suggestion and deployed TS diagram for detecting different water masses alonh the seas through different seasons. They decorated and rearranged the data at lines 213-221 for marking high assembled according to respected water mass in Fig. 3.

Please see the study of Parinos et al., 2017. Cont Shelf Res., 149, 112-123 for an approach on this matter. Please also check if this connection is highlighted/displayed on the clustering of stations in figure 7.

Answer: Authors studied Parinos et al. 2017 thoroughly and visualized TEP assemblages according to density gradient in fig. 5 with necessary details in lines 274-278 and discussions at lines 366, 407-410.

- Section 3.3: I suggest that the authors should consider merging this section with the previous one, sect 3.2, since it is expected that TEP-C distributions will be the same ones as in the case of CTEP since TEP-C=CTEPx0.75 in all cases.

Answer: Authors considered your relevant advice and merged 2.2 with 2.3 at lines 279-290.

- Section 3.6: please check corresponding figure numbers! Fig. 16 is Fig. 6?? Fig. 13 is Fig. 7? Also please follow my comment below ''section 4.1" regarding lines 322-327

Answer: Authors checked and rechecked all numbering of figures in text with necessary correction. They expressed gratitude to the reviewer's humbleness and consideration while reading this manuscript with patience.

**DISCUSSION SECTION**

- Line 337: EPS include also protein-containing Coomassie stainable particles so the term EPS in not equivalent to TEPs only. TEPs are a part of EPS.

Answer: Authors rectified the structure of the sentence as "Study of seasonal trends on EPS (exopolymeric substances i.e. TEP) confirmed the formation of EPS at earlier season in upper sea column with time" at line 331.

- Line 339: resource? Please rephrase

Answer: Rephrased to "phytoplankton cells were reported as precursors of TEP" in line 333.

[Figure]

-Section 4.1: In this section, see also my comment above on lines 322-327, I cannot see the connection between the various phytoplankton species and TEP distributions patterns on Fig.6. Therein the CCA analysis is nicely presented but focused on TEPChla relations and not individual species. I suggest that the authors should consider adding a table outlining TEP-phytoplankton species correlations in support to the discussion introduced in this section or re-phrase the paragraph.

Answer: Authors reconstruct the CCA analysis of TEP with dominant species along all seasons of studied seas with average assemblages for better correspondences analysis in fig. 8. It will help to explain the relations of TEP with different species.

- Line 345: the linkage between TEP and Chl-a in Figs. 6d-3-f is not at all strong in my opinion.

Answer: Authors considered this opinion and rearranged the section 4.1 accordingly.

- Line 348: Same as above for figure 6g.

Answer: Authors changed the numbering with necessary details.

- Lines 374-376: but also, other than sources, to other factors consuming TEPS, i.e. consuming of TEP by bacteria, which is an aspect that cannot be addressed based on the presented dataset.

Answer: Authors tried to mentions these factors with necessary references in lines 335-336.

- Section 4.3: My feeling is that in order to address the potential role of TEP-C in the overall carbon cycle in the study area an essential aspect that is missing from the dataset is considering TEP carbon in respect to POC values. Taking into account the low TEP concentrations recorded overall in the study area, up to 23.2 µg XG eq L-1, and the high chl-a values recorded in summer and moreover the very high chl-a values recorded during winter 2015, TEP-C seems that it probably contributes a very small fraction to POC.

Answer: Authors reconstruct these lines with whole section by mentioning that POC was not measures.

Please see Ortega-Retuerta et al., 2010;2019, Bar-Zeev et al., 2011 and references therein, amongst others, as an example for the relevant discussion. If there are POC data available for the studied samples then they should be in my opinion included in the dataset and discussed in the manuscript.

Answer: Authors studied Ortega-Retuerta et al., 2010;2019 and Bar-Zeev et al., 2011 in detail. They were sited in discussion at 4.3 for relevant TEP-C influences on carbon cycle at lines 458-474.

Overall, I believe that the relevant discussion/ interpretations introduced in section 4.3 are not fully supported by the presented results other than the similarity of TEP distributions with chl-a or nutrients in some cases, which cannot fully support a discussion on TEP role in carbon cycling in the study area

Answer: Authors rearranged this section from TEP-Chl-*a* discussion to TEP-TEP-C relations by seasonal water mass influences at studied seas. Author reproduce a conceptual model as Fig. 10 for the study areas on basis of previous interactions in former researches. It may draw a probable source for TEP concentrations along these study areas.

- Line 426: of the carbon cycle

Answer: Changed in line 443.

- Line 428: the euphotic zone

Answer: Changed in line 446.

- Lines 441-442: zooplankton fecal pellets? This statement is not supported by the presented results.

Answer: Authors removed this part from manuscript.

**TABLES**

Nine tables seem a lot for a manuscript. I suggest that the authors should consider some structural changes as i.e.:

- merging tables 2 and 3 in one major table presenting the concentrations and sinking rates of TEP reported in this study

Answer: Authors merged 2 and 3 with table 1  for seasonal TEP assemblages and TEPs in this study.

- merging tables 4-5-6 and 8 in one shorter major table presenting the concentrations of TEPs available from the literature, considering only the relevant data discussed in the text (lines 380-396)

Answer: Authors merged tables 4-5-6 and 8 in table 2, 3 and 4. Effect of water column on TEP was discussed according to these two tables after dividing into 0-50 m and 50-100 m layered concentration. Table 4 was useful for discussing seasonal TEP concentration at seas.

- incorporating the information provided in tables 7 and 9, especially table 9 were only data from Guo and Sun other than the ones reported in this study are provided, in the main body of the text (sect 4.2)

Answer: Authors merged tables 7 and 9 in table 5 according to this comment, on basis of seasons for having better discussion.

**FIGURES**

Figure 3: Scales used for TEP concentrations should be uniform in all cases, both in min-max values and also scale stepping (step of color change).

Answer: Authors retained Fig. 3 to Fig. 4 by maintaining same colors pattern in scales and max-min values both in integrated and transection view in Fig. 4.

**END OF REVIEW**

---

## Author Comment (AC3) · 28 Apr 2019

Dear Reviewer, I am uploading the supplementary file as your requirements with detail geographical positions of each sampling stations and associated depths. Thank you for your scientific comments.

Please also note the supplement to this comment:
https://www.biogeosciences-discuss.net/bg-2019-58/bg-2019-58-AC3-supplement.pdf

---

## Referee Comment (RC2) · Anonymous Referee #2 · 17 May 2019

This manuscript presents seasonal distributions and sedimentation rates of TEP in the north Chinese sea, and discusses the impact of TEP on carbon cycle in the region.

This manuscript is very poorly written and has to be extensive rewritten. I strongly recommend to have a native English speaker to edit this manuscript prior to any eventual resubmission. At present, it is extremely difficult to read. I do not provide any suggestion for improvement on the style, because at this level, I feel that it is out of the scope and duty of a reviewer to do such an extensive editing. Apart from the low level of English, in my opinion, this manuscript suffers from critical drawbacks regarding the

methods and data interpretation. Below is a list of the main issues.

- The data are presented as average without giving standard deviation. This should be done. All the discussions are based on a comparison of the average values, but based on Figures 4 and 5, there are not differences (no statistical differences), neither between depths, region, nor seasons. Therefore, a paragraph such as "3.4 Seasonal and regional TEP sedimentation", largely overinterpret the data.

- The sampling depths are not given. This should be done, and showed on the surface plots D, 3E and 3F.

- Because TEP concentration is known to vary vertically as a function of the vertical stratification, and to accumulate at the pycnocline, I my opinion it is important to show the vertical stratification at each station during sampling. Although the sampling strategy is not well described (the authors only wrote that "each station was designated with three distinguishing depths for better graphical analysis"), I understand that sampling was performed independently from the position of the pycnocline.

- It is not necessary to show the TEP-C, unless it is to be compared to POC data. Calculating the TEP-C concentration is only useful if one seeks to address the carbon budget.

- Regarding the hydrology of the studied area, the authors wrote that "The Bohai Sea and Yellow Sea had a complex dynamic environment with various seasonal and local geophysical currents". It is not clear if Fig. 2 has been done with data collected during their cruises, or if these data are from a published study. If it is the later, a reference should be given. If it is based on another published work, if it is as dynamic as stated by the authors, how confident can we be that the same scenario occurred during sampling?

- Are the data of the phytoplankton community composition really useful for this study?

- Regarding the search for correspondence between the concentration of TEP and

the other parameters ("3.6 Correspondence relationships of TEP"), the CCA analyses presented in Figure 6 do not show any relationship. Absolutely no pattern emerges from this analysis.

- Last but not least, the approach used to estimate the sedimentation rate of TEP is questionable, as it is does not allow to measure the real TEP sedimentation rate, because this approach assumes that losses from an upper layer is only due to sedimentation. This approach does not take into account other loss processes, such as inner wall attachment, or TEP accumulation at the surface. Such an approach is only valid for determining the sedimentation rate of conventional particles, which only settle down and do not stick to surfaces. As stated by the authors in their introduction, TEP raise at the surface and accumulate in the SML. Therefore, the TEP concentration in each compartment (upper, middle, and bottom) is not only the result of losses from the upper layer towards the bottom layer due to sedimentation, but also the result of the ascent of TEP from the bottom layer towards the surface layer. In addition, all the TEP that may have accumulated in the SML, will not be taken into account in the budget, and may incorrectly be attributed to sinking. In order to validate this approach for studying the sedimentation rate of TEP, one has to be able to close the budget, i.e. to make sure that the sum of the masses of TEP in the 3 compartments equals the initial mass of TEP. If the mass of TEP is not conservative, it is not possible to estimate the actual sedimentation rate.

---

## Author Comment (AC4) · 27 May 2019

Respected Reviewer 2,

Authors are like to response with cordial thanks for these details' scientific comments of reviewer 2. After considering all comments, authors changed the structure of whole manuscript for pursuing a constructive discussion about the main focus of this manuscript according to your suggestions, advises, comments, questions and argues.

Here are the responses to all creative comments accordingly. Please find them in

attachments accordingly. We attached 3 files i.e.

(i) Responses to the comments of Reviewer 2 (ii) Modified and Marked Main Text (iii)Supplementary file 1

A cordial thanks and humble gratitude again for reviewer 2 again.

Please also note the supplement to this comment:
https://www.biogeosciences-discuss.net/bg-2019-58/bg-2019-58-AC4-supplement.pdf

**Supplement:**

**Responses of Reviewer's Comment**

This manuscript presents seasonal distributions and sedimentation rates of TEP in the north Chinese sea, and discusses the impact of TEP on carbon cycle in the region. This manuscript is very poorly written and has to be extensive rewritten. I strongly recommend to have a native English speaker to edit this manuscript prior to any eventual resubmission.

**Responses:** We understand all scientific argues done by respected reviewer and take in account of all these corrections with professional and careful hand. We revised and rephrased the manuscript according to reviewer's requirements. Authors are cordially expressing humble thanks and gratefulness for the time invested by reviewer in this manuscript. We highly appreciated the approaches.

- The data are presented as average without giving standard deviation. This should be done.

**Responses:** Due to the font size (12) we removed the standard deviations. However, we minimized the font size of table and provided necessary standard deviations in required places of table 1.

All the discussions are based on a comparison of the average values, but based on Figures 4 and 5, there are not differences (no statistical differences), neither between depths, region, nor seasons. Therefore, a paragraph such as "3.4 Seasonal and regional TEP sedimentation", largely overinterpret the data.

**Responses:** We merged these sections with previous one for preventing overinterpretation and removed repeated data/figure from manuscript, replacing by TS diagram for interpreting TEP with water masses after following Parinos et all 2017.

- The sampling depths are not given. This should be done, and showed on the surface plots D, 3E and 3F.

**Responses:** The figure has been rectified with depths positions as reviewer's requirements. Sampling depths were given as supplementary data. Previously, due to irregular depths sample, basis on bottom depths; we didn't show their positions in respected figure for their randomness and scattered appearances. As follows:

- Because TEP concentration is known to vary vertically as a function of the vertical stratification, and to accumulate at the pycnocline, I my opinion it is important to show the vertical stratification at each station during sampling. Although the sampling strategy is not well described (the authors only wrote that "each station was designated with three distinguishing depths for better graphical analysis"), I understand that sampling was performed independently from the position of the pycnocline.

**Responses:** Sampling depths were selected on the basis of bottom depths. Different stations possessed different sampling depths as Figure 4. We also draw the aeration of TEP with other biotic parameters along pycnocline in new graph at figure 5. We provided sampling depths with lat-lon as supplementary data to avoid multiple tables in main text.

- It is not necessary to show the TEP-C, unless it is to be compared to POC data. Calculating the TEP-C concentration is only useful if one seeks to address the carbon budget.

**Responses:** We understand your arguments and advises about TEP-C. Hence, authors felt that despite having seasonal dataset of TEP along three seas, it is necessary to project TEP-C data for compering with future data as references if necessary. It will also help the future researchers to calculate and compare TEP-C data with past records.

- Regarding the hydrology of the studied area, the authors wrote that "The Bohai Sea and Yellow Sea had a complex dynamic environment with various seasonal and local geophysical currents". It is not clear if Fig. 2 has been done with data collected during their cruises, or if these data are from a published study. If it is the later, a reference should be given. If it is based on another published work, if it is as dynamic as stated by the authors, how confident can we be that the same scenario occurred during sampling?

**Responses:** Collaboratively modified from previous studies i.e. Hwang et al. 2014; Su 1998; Yuan et al. 2008; Isobe 2008, Zhang et al. 2003, figure 2 was generated. Authors carefully selected the physical circulations along seas from past reports after multiple seasonal comparisons in average for having general discussion. The references were noted at the end of the title of figure 2 as well as in text thoroughly in necessary places.

- Are the data of the phytoplankton community composition really useful for this study?

**Responses:** Due to TEP's close correspondences with dominant species by following Guo and Sun 2018, we provided CCA of TEP with abundant taxa at figure 8 as well as TS assemblages of phytoplankton at figure 5.

- Regarding the search for correspondence between the concentration of TEP and the other parameters ("3.6 Correspondence relationships of TEP"), the CCA analyses presented in Figure 6 do not show any relationship. Absolutely no pattern emerges from this analysis.

**Responses:** We rectified the CCA of TEP with dominant species along all study areas due to its previous reported correspondences with taxa according to Ortega-Retuerta et al. 2010.

- Last but not least, the approach used to estimate the sedimentation rate of TEP is questionable, as it is does not allow to measure the real TEP sedimentation rate, because this approach assumes that losses from an upper layer is only due to sedimentation. This approach does not take into account other loss processes, such as inner wall attachment, or TEP accumulation at the surface. Such an approach is only valid for determining the sedimentation rate of conventional particles, which only settle down and do not stick to surfaces. As stated by the authors in their introduction,

TEP raise at the surface and accumulate in the SML. Therefore, the TEP concentration in each compartment (upper, middle, and bottom) is not only the result of losses from the upper layer towards the bottom layer due to sedimentation, but also the result of the ascent of TEP from the bottom layer towards the surface layer. In addition, all the TEP that may have accumulated in the SML, will not be taken into account in the budget, and may incorrectly be attributed to sinking. In order to validate this approach for studying the sedimentation rate of TEP, one has to be able to close the budget, i.e. to make sure that the sum of the masses of TEP in the 3 compartments equals the initial mass of TEP. If the mass of TEP is not conservative, it is not possible to estimate the actual sedimentation rate.

**Responses:** We have collected samples for TEP and its sinking separately, mentioned at lines 123-125. Authors observed that there is a very close linear relationship among TEP and the summation of TEPs at three compartments. As follows:

[Figure]

Authors discussed the sinking rates in a comparison with physical phenomenon of study areas after considering previously reported TEP's characteristics. We can provide those measured and tabulated records as supplementary data if necessary. We cordially thanking ageing for the comments of reviewer. Hope our answers will satisfy the requirements of reviewer in details.

---

## Author Comment (AC6) · 27 May 2019

Respected Reviewer 2,

I am uploading my modified manuscript here as attachment after necessary changes in structures as your constructive comments. Please find it at the supplementary file section of this replay.

Thank You for your patience.

[Figure]

Please also note the supplement to this comment:
https://www.biogeosciences-discuss.net/bg-2019-58/bg-2019-58-AC6-supplement.pdf
* * *